# Association of Serum Vitamin B_12_ and Circulating Methylmalonic Acid Levels with All-Cause and Cardiovascular Disease Mortality among Individuals with Chronic Kidney Disease

**DOI:** 10.3390/nu15132980

**Published:** 2023-06-30

**Authors:** Shiyi Wu, Wenling Chang, Zhihao Xie, Boshuang Yao, Xiaoyu Wang, Chunxia Yang

**Affiliations:** Department of Epidemiology and Biostatistics, West China School of Public Health and West China Fourth Hospital, Sichuan University, 610041 Chengdu, China; wushiyi510728@gmail.com (S.W.); whitneychang1201@outlook.com (W.C.); hlovml@163.com (Z.X.); boshuangBS@163.com (B.Y.); wangxiaoyu2324@163.com (X.W.)

**Keywords:** chronic kidney disease, mortality, methylmalonic acid, vitamin B_12_, cohort

## Abstract

(1) Background: it is unclear whether serum vitamin B_12_ and circulating methylmalonic acid (MMA) are related with a poor prognosis among individuals with chronic kidney disease (CKD); (2) Methods: this prospective cohort study included 2589 individuals with CKD who participated in the National Health and Nutrition Examination Survey (NHANES) from 1999 to 2004, and from 2011 to 2014, respectively. Hazard ratios (HRs) and 95% Cis for the associations of MMA and vitamin B_12_ levels with the risk of all-cause and cardiovascular disease (CVD) mortality were calculated using multivariable Cox proportional hazards regression models. Restricted cubic spline analyses were used to examine the non-linear association of MMA levels with all-cause and CVD mortality. (3) Results: among the 2589 participants, we identified 1192 all-cause deaths and 446 CVD deaths, respectively, with a median follow-up of 7.7 years. Compared with participants with MMA < 123 nmol/L, those with MMA ≥ 240 nmol/L had an increased all-cause and CVD mortality in the multivariable-adjusted model [HR (95% CI), 2.01 (1.54–2.62) and 1.76 (1.18–2.63), respectively]; (4) Conclusions: higher circulating MMA levels were found to be strongly associated with an elevated all-cause and CVD mortality among individuals with CKD, while serum vitamin B_12_ levels were not associated.

## 1. Introduction

Chronic kidney disease (CKD) is one of the leading noncommunicable causes of death worldwide, affecting more than 10% of the general population in the world, amounting to more than 800 million individuals [1].By 2024, the global burden of CKD has been predicted to increase rapidly and become the 5th leading cause of years of life lost [2,3]. CKD can lead to end-stage renal disease (ESRD), and is associated with an increased risk of cardiovascular morbidity and mortality [4]. Therefore, the identification of modifiable factors related to CKD is crucial to the prevention or delay of CKD-related complications and premature mortality.

Vitamin B_12_, also referred to as cobalamin, is an essential water-soluble vitamin that serves a significant function in metabolism, specifically in DNA synthesis, methylation, and mitochondrial metabolism [5]. Vitamin B_12_ is a cofactor for L-methylmalonyl-coenzyme A mutase, which is involved in the conversion of methylmalonyl-CoA to succinyl-CoA [6]. In vitamin B_12_ deficiency, this process could be impeded by the impaired activity of L-methylmalonyl-CoA mutase, resulting in the conversion of methylmalonyl-CoA to methylmalonic acid (MMA), and then causing the increase in the MMA concentrations in the blood [6]. MMA has been currently recognized as a functional metabolic marker for vitamin B_12_ deficiency [7]. Moreover, methylmalonyl-CoA is an intermediate in the metabolism of propionic acid (PA) [8,9]. In cases where there is no deficiency in vitamin B_12_ or renal dysfunction, elevated levels of MMA may be caused by an increase in the production of PA and its metabolites, or a decrease in their renal clearance [10]. MMA concentrations may be incorrectly increased during renal dysfunction [11,12]. Furthermore, there is increasing evidence in that MMA accumulation has an influence on mitochondrial dysfunction and oxidative stress by regulating the mitochondrial respiratory chain, and initiating the release of reactive oxygen species (ROS) [13,14], which could result in the development and progression of various kidney diseases [15].

The relationship between vitamin B_12_ and mortality has previously been explored in the elderly [16,17], hospitalized [18,19], hemodialysis patients [20], and general population [21]. And the relationship between MMA and mortality has previously been explored in patients with diabetes and the general population [10,22,23]. However, the relationship between MMA and vitamin B_12_ concentrations and mortality in patients with CKD has not been studied. Previous studies have shown that elevated MMA concentrations are associated with a vitamin B_12_ deficiency and renal dysfunction [11,12]. Thus, vitamin B_12_ and MMA levels are markers that may indicate different medical signs and conditions that should be carefully considered in different populations.

In order to fill in gaps in the current research, we conducted a population-based cohort study to investigate the potential association between circulating methylmalonic acid, serum vitamin B_12_ levels, and mortality rates due to all-cause and cardiovascular disease among individuals with chronic kidney disease.

## 2. Materials and Methods

### 2.1. Study Population

The National Health and Nutrition Examination Survey (NHANES) is a survey conducted by the Center for Disease Control and Prevention of the United States. It was designed to assess the health and nutritional status of the non-institutionalized civilian population of all ages in the U.S. The survey uses a stratified, multistage, and probability cluster design. Standardized questionnaires, physical measurements, and laboratory tests are all involved in this survey, and the study design and procedures have been previously published [24]. The National Center for Health Statistics Ethics Review Board approved the study protocol, and all participants provided their informed consent.

In our analyses, we used data from 51,507 participants in five NHANES circles (1999–2000, 2001–2002, 2003–2004, 2011–2012, and 2013–2014, respectively). MMA data were not available in NHANES 2005–2010. CKD was defined as an estimated glomerular filtration rate (eGFR) < 60 mL/min per 1.73 m^2^ or urinary albumin creatinine ratio (ACT) > 30 mg/g, and the eGFR was calculated using the Chronic Kidney Disease Epidemiology Collaboration (CKD-EPI) equation [25]. CKD was classified into 5 stages according to the Kidney Disease: Improving Global Outcomes (KDIGO) guideline: stage 1 (eGFR ≥ 90 mL/min/1.73 m^2^ with ACT > 30 mg/g), stage 2 (eGFR > 60 and <90 with ACT > 30 mg/g), stage 3 (eGFR > 30 and <60), stage 4 (eGFR > 15 and <30), and stage 5 (eGFR < 15 mL/min/1.73 m^2^), respectively [26]. We excluded participants without CKD (*n* = 46,887), those <20 years (*n* = 962), those with pregnancy (*n* = 17), those with missing MMA values (*n* = 38), those with missing vitamin B_12_ values (*n* = 5), and those with a lack of covariates (*n* = 1009) including education, marital status, poverty-to-income ratio (PIR), body mass index (BMI), smoking, drinking, physical activity, Healthy Eating Index-2015 (HEI-2015), systolic pressure, and eGFR. In total, 2589 participants with CKD were included and available for analyses (Figure 1).

### 2.2. Measurements of MMA and Vitamin B_12_


MMA was measured in the plasma and/or serum, with plasma being preferred. MMA was measured in the NHANES from 1999 to 2004, respectively, using a gas chromatography-mass spectrometry (GC/MS) procedure that required 275 μL of the sample and had a low throughput (36 samples/run). In the NHANES 2011–2012 circle, MMA was measured using liquid chromatography-tandem mass spectrometry (LC-MS/MS) with multiple reaction monitoring. These two measurement methods had an excellent comparability and bias-free agreement [27]. The units used for MMA in this study were nmol/L.

In the NHANES III and 1999–2016 circles, the Bio-Rad Laboratories Quantaphase II radioimmunoassay was used to measure vitamin B_12_, while in the NHANES 2011–2014 the fully automated Roche electrochemiluminescence immunoassay was. But the weighted distributions of the 2005–2006 and 2011–2012 datasets were slightly different, resulting in the Deming regression equations [Roche vitamin B_12_ = 10 × (0.97 × log10 (Bio-Rad vitamin B_12_) + 0.14 pg/mL)] being recommended to use for our analyses [28]. The vitamin B_12_ in pg/mL was converted into pmol/L by multiplying it with 0.738. In our study, the vitamin B_12_ values were converted in the Roche assay to pmol/L.

### 2.3. Ascertainment of Mortality

The endpoint in our study was all-cause and CVD mortality. Death data were obtained by linking the public-use database from the National Death Index (NDI) up through 31 December 2019. Causes of death were determined using the International Classification of Diseases Tenth Revision code (ICD-10) codes. All-cause mortality was defined as any cause of death, and CVD mortality was coded as I00–I09, I11, I13, I20–I51, and I60–I69. Participants with no record of death during follow-up were considered to be alive.

### 2.4. Measurement of Covariates

Data regarding age, sex (male and female), race (Hispanic, non-Hispanic white, non-Hispanic black, and others), education (less than high school, high school, and college or higher), marital status (married, separated, and never married), PIR (≤1, 1–4, and ≥4) [29], smoking (never, former, and current), and drinking (none, moderate, and heavy), physical activity (low, moderate, and high) [30] were all collected from the baseline standardized questionnaires. BMI was calculated as the weight (kg) divided by the height squared (m^2^). HEI-2015 is a scoring system that ranges from 0 to 100 and is based on the 24 h dietary recalls which were conducted by trained interviewers. This system consists of 13 components, including eight food groups and five nutrients [31]. A higher score indicates a higher quality of diet. Systolic blood pressures were calculated as the mean of the readings obtained according to a standardized protocol.

### 2.5. Statistical Analysis

Statistical analyses were performed using R version 4.2.2, Stata version 15.1. All estimates were weighed with survey design, survey non-response, and post-stratification, which were considered in all analyses according to the recommendations from the NHANES Analytic and Reporting Guidelines Baseline [32]. Statistical tests were two-tailed, and the statistical significance was defined as *p* < 0.05.

In this study, the baseline characteristics of the total population and quartiles of the baseline MMA and vitamin B_12_ levels were presented. This comparison was performed using the Wald and F test for continuous variables, and the Rao–Scott chi-square test for categorical variables. The mean (standard error, SE) was used to express the continuous variables, while counts (percentage, %) were used to express the categorical variables

In this study, hazard ratios (HRs) and 95% confidence intervals (Cis) were calculated using multivariable Cox proportional hazards regression models to determine the associations between the MMA and vitamin B_12_ levels and the risks of all-cause and CVD mortality. Three multivariable-adjusted models were constructed stepwise to account for potential confounding. In Model 1, we adjusted for the NHANES circles (categorical variables for 1999–2000, 2001–2002, 2003–2004, 2011–2012, and 2013–2014, respectively), age, and sex. In Model 2, we additionally adjusted for race, education, marital status, PIR, BMI, smoking, drinking, physical activity, diet quality, systolic blood pressure, and eGFR. In Model 3, we further adjusted for the vitamin B_12_ and MMA levels.

To examine the non-linear association of the natural logarithm of the MMA levels with mortality, restricted cubic spline analyses with 4 knots (5th, 35th, 65th, and 95th percentiles, respectively) were conducted. These analyses were limited to values between the 5th and 95th percentiles to minimize the influence of potential outliers, with the 25th percentile as the reference point, respectively.

We further stratified these analyses by age (<65 and ≥65 years), sex (male and female), BMI (<30 and ≥30 kg/m^2^), smoking (never, former, and current), drinking (none, moderate and heavy), physical activity (low, moderate, and high), eGFR (≥60 and <60 mL/min/1.73 m^2^), and CKD stages (Stage 1, 2, and 3–5). The interaction (modification effect) between a stratifying variable and the composite MMA quartiles were estimated by adding interaction terms into the Cox proportional hazard regression and were assessed using the Wald test.

We performed three sensitivity analyses: (1) participants with a history of CVD and diabetes were further excluded from the main analyses. (2) Participants who died within 3 years of follow-up were excluded from the main analyses. (3) We further adjusted for several biomarkers, including homocysteine, HbA_1c_, and total cholesterol.

## 3. Results

### 3.1. Baseline Characteristics

Of the 2589 study participants aged 59.0 years (SE: 0.6), 1324 (46.8%) were men, and we identified 1192 all-cause deaths and 446 CVD deaths with a median follow-up of 7.7 years, and with a median follow-up of 14.5 years, 13.0 years, 12.2 years, 7.7 years, and 5.8 years, respectively, in five different circles (1999–2000, 2001–2002, 2003–2004, 2011–2012, and 2013–2014, respectively). For the analysis of MMA, participants with higher MMA levels were more likely to be older, non-Hispanic white, married, non-smokers, high physical activity, higher diet quality, higher systolic blood pressure, worsened CKD stage, lower vitamin B_12_ and eGFR (Table 1). For the analysis of vitamin B_12_, participants with higher vitamin B_12_ levels were more likely to be older, female, non-Hispanic white, lower BMI, higher diet quality, higher systolic blood pressure, lower eGFR, and lower MMA levels (Appendix A).

### 3.2. MMA and Mortality

Restricted cubic splines presented no evidence for non-linear associations between the MMA levels and all-cause mortality (*p* for non-linearity = 0.385), nor with CVD mortality (*p* for non-linearity = 0.789) (Figure 2). After multivariable adjustment, the association showed a dose-response pattern, with a 30% increased risk of CVD mortality per unit increase in log-transformed MMA (HR, 1.30; 95% CI: 1.08, 1.57). When analyzing categorical MMA, compared with the reference group (first quartile, MMA < 123 nmol/L), the HRs (95% CI) of all-cause mortality with MMA 123–169 nmol/L, MMA 170–239 nmol/L, and MMA ≥ 240 nmol/L were 1.20 (0.91, 1.57), 1.28 (0.94, 1.76), and 2.01 (1.54, 2.62), respectively. Similarly, compared with the reference group (the first quartile), the HRs (95% CI) for CVD mortality were 1.29 (0.85, 1.97) for the second quartile, 1.04 (0.69, 1.58) for the third quartile, and 1.76 (1.18, 2.63) for the fourth quartile, respectively (Table 2).

### 3.3. Vitamin B_12_ and Mortality

Compared to those with vitamin B_12_ < 284.1 pmol/L, the HRs (95%) of all-cause mortality in participants with vitamin B_12_ 284.1–397.1 pmol/L and 397.2–534.5 pmol/L, ≥534.6 pmol/L were 1.01 (0.78, 1.31), 0.93 (0.71, 1.22), and 1.22 (0.93, 1.61), respectively. Compared with the reference group (the first quartile), the HRs (95% CI) for CVD mortality were 1.02 (0.74, 1.39) for the second quartile, 1.02 (0.72, 1.44) for the third quartile, and 1.26 (0.94, 1.69) for the fourth quartile, respectively (Appendix A).

### 3.4. Stratified and Sensitivity Analyses

In stratification analyses, the dose–response relationship between MMA and all-cause mortality was largely stratified by age (<65 and ≥65 years), sex (male and female), BMI (<30 and ≥30 kg/m^2^), smoking (never or former, and current), drinking (none, moderate, and heavy), physically active (low, moderate, and high), eGFR (<60 and ≥60 mL/min/1.73 m^2^), and CKD stages (stage1, 2, and 3–5) (all *p* ≥ 0.128 for interaction) (Table 3). Similarly, after correction for multiple testing, there were no significant interactions found between the levels of MMA and the risk of CVD mortality (Appendix A).

In general, the results were robust to sensitivity analyses when excluding the participants with a history of CVD and diabetes (Appendix A), participants who died within 3 years of follow-up (Appendix A), and adjusting for several biomarkers, including homocysteine, HbA_1_c, and total cholesterol (Appendix A).

## 4. Discussion

This study is the first to analyze the levels of circulating MMA and vitamin B_12_ in individuals with CKD and their potential risks for all-cause and CVD mortality. No evidence of a non-linear association was found between the MMA levels and all-cause or CVD mortality in our study. It could be inferred that the relationship between the MMA levels and mortality is probably closer to a linear relationship. Our findings demonstrated that participants with MMA levels ≥ 240 nmol/L are at a higher risk for all-cause and CVD mortality compared to those with MMA levels < 123 nmol/L. Cox regression analysis of categorical MMA levels following the restricted cubic spline analysis of continuous MMA levels could reduce the effect of non-linear patterns in the data in the Cox regression model and improve the predictive power of the Cox regression model. Such associations remained robust after a series of sensitivity analyses. However, no significant association was found between serum vitamin B_12_ levels and the mortality risk.

The association between vitamin B_12_ levels and mortality has been controversial in epidemiological studies. A study conducted in the Netherlands indicated that higher levels of plasma vitamin B_12_ were associated with an increased risk of all-cause mortality, even after adjusting for age, sex, renal function, and other clinical and laboratory variables [33], but a NHANES study found that although there was a small but significant increase in CVD mortality in both low and high serum B_12_ groups, the overall association of the serum B_12_ concentration with mortality is U-shaped. These results do not support the suggestion that high serum B_12_ concentrations are inherently harmful or detrimental [21]. Studies on the association between vitamin B_12_ levels and mortality in older people have yielded conflicting results. The Newcastle 85+ study found that a higher concentration plasma vitamin B_12_ in women was linked to an increased risk of all-cause and CVD mortality [17], while an Australian study of people aged 55 years and older did not find any association between serum vitamin B_12_ and CHD and all-cause mortality [34]. Similarly, a community-dwelling Irish study showed that Vitamin B_12_ levels were not associated with death rates [35]. In populations with chronic diseases, studies have shown that both low and high serum levels of vitamin B_12_, as well as low serum levels of folate, are significantly associated with a higher risk of CVD mortality among individuals with T2D [36]. Additionally, vitamin B_12_ levels have been identified as an independent predicting factor for the three month mortality rate in AoCLF patients [37]. However, a cohort study of 1684 ICU patients found no significant associations between serum B_12_ levels and mortality after adjustment for liver function and liver disease [18]. While the evidence does not fully support considering vitamin B_12_ alterations as reliable risk markers for cardiovascular mortality in CKD and ESRD populations [38], there is little evidence available on the vitamin B_12_ status and long-term health outcomes.

Multiple epidemiological studies have found that MMA may be an independent risk factor for predicting mortality. A study conducted on the general population from the NHANES revealed that a high level of mitochondrial-derived MMA was strongly associated with an increased all-cause and cardiovascular mortality [23]. Another prospective study conducted on two cohorts demonstrated that elevated MMA levels were associated with a higher risk of acute myocardial infarction (AMI) and mortality in patients with a suspected or confirmed CHD [39]. MMA is a potential risk factor of CVD, diabetic complications, and dementias, and higher MMA levels have been associated with all-cause mortality, cancer mortality, and CVD mortality among individuals with diabetes [10,22]. Similar results were confirmed in our study, where there was a significant association found between the MMA levels and all-cause and cardiovascular mortality. It may suggest, therefore, that the levels of circulating MMA can predict long-term all-cause and cardiovascular mortality in patients with CKD.

MMA is currently best known for its use as a functional marker for vitamin B_12_ deficiency [7]. This deficiency can impair the activity of L-methylmalonyl-CoA mutase, resulting in the conversion of methylmalonyl-CoA to MMA instead of succinyl-CoA [6]. But vitamin B_12_ deficiency can only explain part of the increase in the MMA levels observed in adults [40,41]. Mitochondrial dysfunction [42,43] and impaired renal function [44,45,46] may account for proportions of the elevated MMA levels. Previous studies have suggested that MMA levels may be influenced by various factors, including catabolism, dietary components, and gut microbial production [10]. Mitochondrial dysfunction, characterized by impaired bioenergetics and a redox imbalance, is prevalent in patients with CKD [47] and many chronic diseases, such as CVD and diabetes [48]. MMA has been found to play a significant role in causing mitochondrial dysfunction and oxidative stress, both in vitro and in vivo, and does so by disrupting the mitochondrial respiratory chain and inducing reactive oxygen species generation [13,14]. This may then suggest that the underlying mechanism between the high levels of MMA and the risk of death may be related to mitochondrial dysfunction leading to disease progression, characterized by impaired bioenergetics and a redox imbalance in patients with CKD [23]. Patients with CKD may experience changes in the composition and metabolic activity of their gut microbiota due to their medications and diet. Elevated levels of MMA may indicate high levels of high PA bacterial abnormalities in the gut (MMA is an intermediate in the metabolism of PA), and the microbiota may impact the overall health by secreting metabolites associated with insulin resistance, obesity, endothelial dysfunction, and cardiovascular aging [49], all of which may also account for the increased risk of death in individuals through high levels of MMA by affecting catabolism. Furthermore, the association of MMA with all-cause and cardiovascular mortality risk remained significant even after adjusting for vitamin B_12_ and eGFR, indicating that this association is independent of the traditional risk factors.

There was no significant predictive effect observed between the serum vitamin B_12_ levels with CKD mortality. This could be because fewer outcome events occurred in our study, lacking the statistical power to detect associations between vitamin B_12_ levels and mortality. In addition, vitamin B_12_ exhibits a small effect on MMA levels, which may explain its non-significant association with mortality.

Thus, MMA is likely to be an underestimated biomarker that can be used to predict poor prognoses in individuals with CKD, and to clarify the role of MMA and vitamin B_12_ in long-term health, further mechanistic studies are needed.

As far as strengths of our study, we used a nationally large representative sample to assess the association of MMA and vitamin B_12_ levels with all-cause and CVD mortality among individuals with CKD, adjusting several potentially confounding factors, and assessing the robustness of our findings using several sensitivity analyses, which could facilitate the generalization of these findings. However, our study had several limitations that should be considered. First, MMA and vitamin B_12_ levels were determined based on a single measurement, which may not accurately reflect the long-term status of the participants. Second, both MMA and vitamin B_12_ were analyzed using different analytical methods at different stages in the NHANES. Although attempts have been made to statistically harmonize the values, the results cannot be as good as those obtained using the same laboratory methods at the same time. Third, we were not able to distinguish between the inborn and postnatal increases in MMA and vitamin B_12_. However, the prevalence of hereditary methylmalonic acidemia [50] and CblC deficiency [51] are pretty low, meaning they would not have as much of an impact on our conclusions. Fourth, due to the observational study design we utilized, our results are unable to completely eliminate the possibility of unascertained confounding factors. Thus, further studies are necessary to validate our findings. We cannot determine the causal relationship from this present study. Last, we categorized classified MMA and vitamin B_12_ levels according to the quartiles, meaning our results may not be consistent with those of other studies using different cut points.

## 5. Conclusions

Our study found that higher circulating levels of MMA were associated with an increased risk of all-cause and CVD mortality. However, we did not observe a significant association between serum vitamin B_12_ levels and mortality risk.

## Figures and Tables

**Figure 1 nutrients-15-02980-f001:**
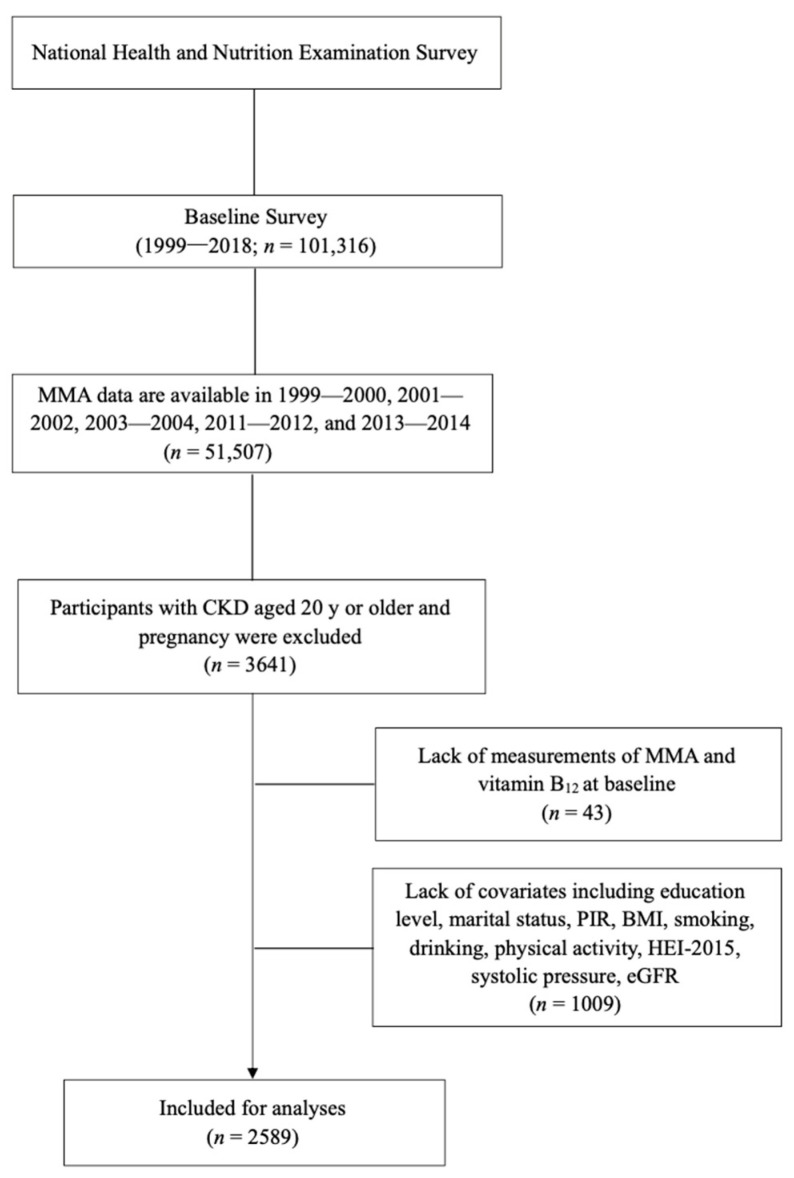
Flowchart of the participants included in the analyses.

**Figure 2 nutrients-15-02980-f002:**
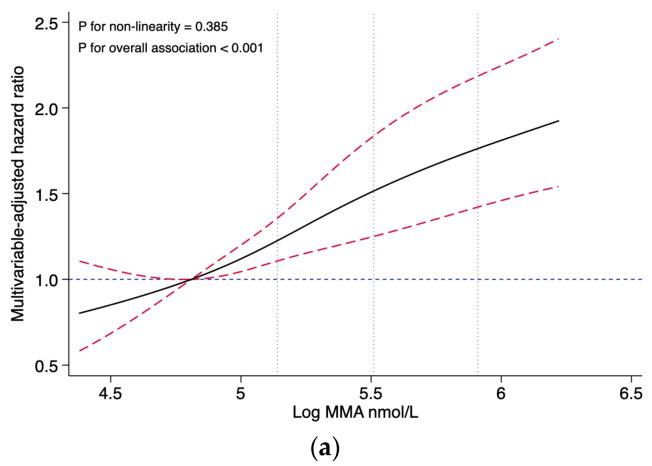
Association of the MMA levels with all-cause and CVD mortality, evaluated using restricted cubic splines. (**a**) Association of the MMA levels with all-cause mortality. (**b**) Association of the MMA levels with CVD mortality. Abbreviations: MMA, methylmalonic acid. Hazard ratios and confidence intervals were estimated for MMA using four knots placed at specific percentiles of the natural logarithm of MMA. The reference value was set at the 25th percentile, and the dotted vertical lines represent the 50th, 75th, and 90th percentiles, respectively. The results are presented using solid black lines for the hazard ratios, red dashed lines (indicating the 95% confidence interval) for the confidence intervals and blue dashed lines for hazard ratios for reference lines of 1.0. The model was adjusted for the NHANES circles, age (continuous, year), sex (male and female), race (Hispanic, non-Hispanic white, non-Hispanic black, and others), education (less than high school, high school, and college or higher), marital status (married, separated, and never married), PIR (≤1, 1–4, and ≥4), BMI (continuous, kg/m^2^), smoking (never, former, and current), drinking (never, former, and current), physical activity (low, moderate, and high), diet quality (continuous, HEI-2015 score), systolic blood pressure (continuous, mmHg), eGFR (continuous, mL/min/1.73 m^2^), and vitamin B_12_ (continuous, pmol/L). No non-linear association was observed between MMA and all-cause mortality (*p* for non-linearity = 0.385), and no non-linear association was observed between MMA and CVD mortality (*p* for non-linearity = 0.789).

**Table 1 nutrients-15-02980-t001:** Baseline characteristics of the study participants for CKD among U.S. adults by their baseline MMA levels.

Characteristic	Participants, No. (%)					*p*-Value ^a^
	MMA Levels, nmol/L					
	Total	Quartile 1 (<123)	Quartile 2 (123–169)	Quartile 3 (170–239)	Quartile 4 (≥240)	
**Participants, No. (%)**	2589 (100.0)	642 (24.8)	592 (22.8)	665 (25.7)	690 (26.7)	-
**Age, years, mean (SE)**	59.04 (0.6)	46.79 (0.8)	57.75 (1.1)	63.64 (0.9)	67.55 (0.8)	<0.001
**Sex, *n* (%)**						0.095
Male	1324 (46.8)	294 (42.2)	306 (47.1)	360 (51.0)	364 (46.6)	
Female	1265 (53.2)	348 (57.8)	286 (52.9)	305 (49.0)	326 (53.4)	
**Race, *n* (%)**						<0.001
Hispanic	580 (12.6)	223 (23.6)	124 (10.8)	125 (9.4)	108 (7.1)	
Non-Hispanic white	1302 (69.7)	190 (52.2)	287 (70.1)	389 (76.5)	436 (79.6)	
Non-Hispanic black	543 (11.5)	189 (17.8)	135 (12.4)	121 (9.0)	98 (7.1)	
Others	164 (6.1)	40 (6.4)	46 (6.8)	30 (5.1)	48 (6.3)	
**Education, *n* (%)**						0.179
Less than high school	911 (25.2)	226 (23.8)	203 (24.8)	239 (25.0)	243 (27.0)	
High school	594 (24.2)	146 (22.4)	133 (22.7)	141 (23.9)	174 (27.8)	
College or higher	1084 (50.7)	270 (53.9)	256 (52.6)	285 (51.1)	273 (45.2)	
**Marital status, *n* (%)**						<0.001
Married	1337 (53.9)	330 (50.5)	320 (55.6)	354 (58.2)	333 (51.1)	
Separated	980 (34.4)	204 (30.6)	204 (31.7)	260 (34.2)	312 (40.6)	
Never married	272 (11.9)	108 (18.9)	68 (12.8)	51 (7.7)	45 (8.4)	
**PIR, *n* (%)**						0.089
≤1	554 (17.0)	154 (20.4)	125 (16.2)	139 (15.5)	136 (16.1)	
1 to 4	1512 (54.9)	365 (53.5)	351 (53.5)	366 (52.6)	430 (59.9)	
≥4	523 (28.1)	123 (26.1)	116 (30.4)	160 (31.9)	124 (23.9)	
**Smoking, *n* (%)**						<0.001
Never	2043 (76.9)	463 (69.1)	448 (73.6)	548 (80.6)	584 (83.8)	
Former	68 (3.1)	23 (5.2)	14 (2.3)	17 (2.9)	14 (2.1)	
Current	478 (20.0)	156 (25.8)	130 (24.0)	100 (16.5)	92 (14.0)	
**Drinking, *n* (%)**						0.202
None	900 (33.3)	208 (30.5)	195 (30.1)	230 (33.2)	267 (39.1)	
Moderate	1492 (58.2)	374 (59.8)	342 (60.4)	387 (58.2)	389 (54.4)	
Heavy	197 (8.6)	60 (9.7)	55 (9.6)	48 (8.6)	34 (6.5)	
**Physical activity, *n* (%)**						<0.001
Low	1173 (49.1)	339 (58.8)	291 (53.6)	292 (46.4)	251 (37.8)	
Moderate	549 (21.3)	135 (20.2)	118 (20.2)	149 (23.0)	147 (21.8)	
High	867 (29.6)	168 (21.0)	183 (26.2)	224 (30.6)	292 (40.4)	
**CKD stages**						<0.001
Stage 1	938 (41.1)	458 (74.7)	253 (47.9)	143 (27.5)	84 (15.4)	
Stage 2	668 (24.2)	135 (17.4)	187 (27.1)	205 (30.8)	141 (21.2)	
Stage 3	908 (32.8)	48 (7.8)	150 (24.9)	312 (41.1)	398 (56.3)	
Stages 4 and 5	75 (2.0)	1 (0.1)	2 (0.1)	5 (0.6)	67 (7.2)	
**BMI, kg/m^2^, mean (SE)**	29.43 (0.2)	29.51 (0.4)	29.32 (0.4)	29.41 (0.3)	29.49 (0.3)	0.060
**HEI-2015, mean (SE)**	53.08 (0.4)	50.88 (0.7)	53.28 (0.6)	53.82 (0.7)	54.28 (0.6)	<0.001
**Systolic blood pressure, mm Hg, mean (SE)**	133.42 (0.5)	127.90 (1.1)	134.89 (1.1)	134.81 (1.2)	135.94 (1.2)	<0.001
**eGFR, mL/min/1.73 m^2^, mean (SE)**	81.42 (0.7)	105.16 (1.3)	86.74 (1.4)	73.87 (1.2)	60.77 (1.3)	<0.001
**Vitamin B_12_, pmol/L, mean (SE)**	481.77 (16.5)	556.99 (50.8)	493.10 (22.0)	449.76 (11.6)	430.01 (36.0)	<0.001

Abbreviations: BMI, body mass index; CKD, chronic kidney disease; eGFR, estimated glomerular filtration rate; HEI-2015, Healthy Eating Index-2015; MMA, methylmalonic acid; PIR, family poverty-to-income ratio; and SE, stand error. ^a^ *p* values were calculated using the Wald and F test for the continuous variables, and the Rao–Scott chi-square test for the categorical variables.

**Table 2 nutrients-15-02980-t002:** Association of the MMA levels with all-cause mortality and CVD mortality among U.S. adults with CKD.

Cause of death	MMA Levels (nmol/L), HR (95% CI)					*p*-Value for Trend ^a^
	Log (MMA)	Quartile 1 (<123)	Quartile 2 (123–169)	Quartile 3 (170–239)	Quartile 4 (≥240)	
**All-cause mortality**						
No. of cases/person-years	1192/24,301	169/7670	233/5888	337/5851	453/4892	
Model 1	1.49 (1.29, 1.73)	1.00 [Reference]	1.16 (0.89, 1.51)	1.22 (0.91, 1.62)	1.96 (1.54, 2.50)	<0.001
Model 2	1.50 (1.29, 1.75)	1.00 [Reference]	1.20 (0.91, 1.57)	1.28 (0.94, 1.76)	2.01 (1.54, 2.61)	<0.001
Model 3	1.09 (0.90, 1.34)	1.00 [Reference]	1.20 (0.91, 1.57)	1.28 (0.94, 1.76)	2.01 (1.54, 2.62)	<0.001
**CVD mortality**						
No. of cases/person-years	446/24,301	61/7670	95/5888	115/5851	175/4892	
Model 1	1.39 (1.15, 1.68)	1.00 [Reference]	1.28 (0.86, 1.89)	1.06 (0.74, 1.51)	1.94 (1.38, 2.74)	<0.001
Model 2	1.30 (1.08, 1.57)	1.00 [Reference]	1.29 (0.84, 1.97)	1.04 (0.69, 1.58)	1.76 (1.18, 2.63)	<0.001
Model 3	1.30 (1.08, 1.57)	1.00 [Reference]	1.29 (0.85, 1.97)	1.04 (0.69, 1.58)	1.76 (1.18, 2.63)	<0.001

Abbreviations: CI, confidence interval; CKD, chronic kidney disease; CVD, cardiovascular disease; HR, hazard ratio; and MMA, methylmalonic acid. Model 1: adjusted for the NHANES circles, age (continuous, year), and sex (male and female). Model 2: further adjustment for race (Hispanic, non-Hispanic white, non-Hispanic black, and others), education (less than high school, high school and college or higher), marital status (married, separated, and never married), PIR (≤1, 1–4, and ≥4), BMI (continuous, kg/m^2^), smoking (never, former, and current), drinking (never, former, and current), physical activity (low, moderate, and high), diet quality (continuous, HEI-2015 score), systolic blood pressure (continuous, mmHg), eGFR(continuous, mL/min/1.73 m^2^) based on Model 1. Model 3: further adjustment for vitamin B_12_ (continuous, pmol/L) based on Model 2. ^a^ *p* values for trend were assessed using the median level of each quartile MMA levels and modeling it as a continuous variable.

**Table 3 nutrients-15-02980-t003:** Subgroup analyses for the HRs of the MMA levels and all-cause mortality among U.S. adults with CKD.

Subgroup	MMA levels (nmol/L), HR (95% CI)					*p*-Value for Interaction ^a^
	Log (MMA)	Quartile 1 (<123)	Quartile 2 (123–169)	Quartile 3 (170–239)	Quartile 4 (≥240)	
**Age, years**						0.255
<65	1.93 (1.50, 2.47)	1.00 [Reference]	1.39 (0.95, 2.06)	1.84 (1.24, 2.73)	2.83 (1.81, 4.43)	
≥65	1.34 (1.20, 1.50)	1.00 [Reference]	1.02 (0.80, 1.30)	1.19 (0.94, 1.50)	1.75 (1.38, 2.21)	
**Sex**						0.536
Male	1.33 (1.17, 1.51)	1.00 [Reference]	1.08 (0.82, 1.42)	1.21 (0.92, 1.58)	1.73 (1.31, 2.29)	
Female	1.69 (1.42, 2.01)	1.00 [Reference]	1.25 (0.91, 1.72)	1.58 (1.15, 2.17)	2.33 (1.69, 3.22)	
**BMI, kg/m^2^**						0.764
<30	1.26 (1.11, 1.42)	1.00 [Reference]	1.08 (0.84, 1.39)	1.32 (1.02, 1.70)	1.68 (1.30, 2.17)	
≥30	2.05 (1.67, 2.51)	1.00 [Reference]	1.15 (0.81, 1.64)	1.28 (0.90, 1.82)	2.40 (1.68, 3.45)	
**Smoking**						0.325
Never or former	1.44 (1.29, 1.59)	1.00 [Reference]	1.17 (0.93, 1.48)	1.39 (1.11, 1.75)	1.99 (1.58, 2.51)	
Current	1.67 (1.20, 2.34)	1.00 [Reference]	1.29 (0.80, 2.08)	1.44 (0.87, 2.36)	2.48 (1.41, 4.36)	
**Drinking**						0.886
None	1.45 (1.24, 1.69)	1.00 [Reference]	1.12 (0.78, 1.60)	1.47 (1.03, 2.09)	2.15 (1.51, 3.06)	
Moderate	1.45 (1.26, 1.67)	1.00 [Reference]	1.14 (0.86, 1.50)	1.28 (0.97, 1.69)	1.85 (1.39, 2.47)	
Heavy	1.84 (1.01, 3.33)	1.00 [Reference]	1.19 (0.55, 2.60)	1.15 (0.51, 2.58)	2.50 (1.03, 6.11)	
**Physical activity**						0.128
Low	1.46 (1.24, 1.72)	1.00 [Reference]	0.96 (0.70, 1.33)	1.20 (0.88, 1.65)	1.90 (1.37, 2.65)	
Moderate	1.75 (1.36, 2.25)	1.00 [Reference]	1.51 (0.93, 2.44)	1.98 (1.23, 3.19)	2.96 (1.77, 4.96)	
High	1.33 (1.14, 1.56)	1.00 [Reference]	1.16 (0.83, 1.63)	1.20 (0.85, 1.69)	1.80 (1.27, 2.54)	
**eGFR, mL/min/1.73 m^2^**						0.614
<60	1.31 (1.13, 1.52)	1.00 [Reference]	1.02 (0.65, 1.59)	1.36 (0.90, 2.05)	1.74 (1.15, 2.62)	
≥60	1.44 (1.26, 1.66)	1.00 [Reference]	1.22 (0.96, 1.54)	1.34 (1.04, 1.73)	2.17 (1.66, 2.84)	
**CKD stage**						0.147
1	1.29 (1.02, 1.64)	1.00 [Reference]	1.07 (0.74, 1.54)	1.49 (0.99, 2.24)	1.92 (1.18, 3.14)	
2	1.71 (1.41, 2.08)	1.00 [Reference]	1.37 (0.99, 1.89)	1.35 (0.96, 1.91)	2.44 (1.73, 3.46)	
Stage 3–5	1.31 (1.13, 1.52)	1.00 [Reference]	1.02 (0.65, 1.59)	1.36 (0.90, 2.05)	1.74 (1.15, 2.62)	

Abbreviations: BMI, body mass index; CI, confidence interval; CKD, chronic kidney disease; eGFR, estimated glomerular filtration rate; HR, hazard ratio; and MMA, methylmalonic acid. Model was adjusted for the NHANES circles, age (continuous, year), sex (male and female), race (Hispanic, non-Hispanic white, non-Hispanic black, and others), education (less than high school, high school, and college or higher), marital status (married, separated, and never married), PIR (<1, 1–4, >4), BMI (continuous, kg/m^2^), smoking (never or former, and current), drinking (never, former, and current), physical activity (low, moderate, and high), diet quality (continuous, HEI-2015 score), systolic blood pressure (continuous, mmHg), eGFR(continuous, mL/min/1.73 m^2^), and vitamin B_12_ (continuous, pmol/L). ^a^ *p* for interaction was estimated by including an interaction term of the composite MMA and stratifying variables in the Cox regression model and assessing using the Wald test.

## Data Availability

Publicly available datasets were analyzed in this study. These data can be found at https://www.cdc.gov/nchs/nhanes (accessed on 1 November 2021).

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
