# Peer review of "Association of Serum Vitamin B12 and Circulating Methylmalonic Acid Levels with All-Cause and Cardiovascular Disease Mortality among Individuals with Chronic Kidney Disease"

_nutrients, 2023, doi:10.3390/nu15132980_

Round 1

Reviewer 1 Report

This is a potentially interesting paper to verify the association between of methylmalonic acid (MMA) and vitamin B12 levels with risk of all-cause and CVD mortality using NHANES 1999 to 2004 and from 2011 to 2014 data. This study intends to continue the study published by Herrmann W et al. in Clin Chem Lab Med 2001; 39(8):739–746. In this paper, only the renal function is changed to CKD, and the relationship between MMA and vitamin B12 and CKD is investigated for all-cause and CVD mortality Relationship. However, major modifications are required.

1.     This study is a stratified, multistage, and probability cluster design, and Cox proportional regression was used to analyze the data. Authors should describe the follow-up situation and length of follow-up of CKD patients recruited in different years of the study design. However, in the fourth point of research limitations (line 323), it says that we cannot establish causality from the current study. This is very confusing to me! A follow-up study using Cox proportional regression to analyze the data and saying that causality cannot be established is very contradictory. Authors must explain their rationale well.

2. The full name should appear on the first occurrence of the abbreviation in the text, e.g. HEI-2015 on line 84. What is this score? How this score was obtained?

 3. The titles of “a” and “b” in Figure 1 are the same, are they wrong? The Discussion section did not discuss the non-linear association (Figure 1), what is its analytical significance? It should be explained how it differs from the results of the Cox proportion hazard model analysis.

4. Line 138, Should the Wald F-test be the Wald and F test?

5. The method of the interaction analyses in the subgroup analyzes in Table 3 should be described in detail in 2.5 Statistical Analyses. Not just rendered in footnote.

6. This study did not observe the correlation between vitamin B12 and all cause and CVD mortality. Inconsistent results from many studies were also cited in the Discussion. Should the authors try to explain possible reasons why no correlation was observed in this study?

Reviewer 2 Report

So an association was found but why? The discussion section needs an explanation as to why there might be an association between circulating MMA and mortality in individuals with CKD. What are some possible mechanisms? Why would MMA be a better predictor of mortality than serum B12? 

Lines 308-312: The 2 sentences are somewhat confusing and contradictory. Are there other factors? What are they? What factors have been identified by which authors?

Line 321: Not clear what is meant by "inborn versus postnatal increases" in MMA and B12.

Another study weakness is the fact that both B12 and MMA were analyzed by different analytical methods at different time phases of the NHANES study. Although attempts were made to statistically harmonize the values, the results cannot be as good as measurements done using the same laboratory methods and at the same time.
